# Two-Day Cardiopulmonary Exercise Testing in Females with a Severe Grade of Myalgic Encephalomyelitis/Chronic Fatigue Syndrome: Comparison with Patients with Mild and Moderate Disease

**DOI:** 10.3390/healthcare8030192

**Published:** 2020-06-30

**Authors:** C (Linda) M. C. van Campen, Peter C. Rowe, Frans C. Visser

**Affiliations:** 1Stichting Cardio Zorg, 2132 HN Hoofddorp, The Netherlands; fransvisser@stichtingcardiozorg.nl; 2Department of Paediatrics, John Hopkins University School of Medicine, John Hopkins University, Baltimore, MD 21218, USA; prowe@jhmi.edu

**Keywords:** chronic fatigue syndrome, cardiopulmonary exercise testing, oxygen consumption, VO_2_ peak, ventilatory threshold, VO_2_ VT, myalgic encephalitis, workload, ME/CFS severity grade

## Abstract

Introduction: Effort intolerance along with a prolonged recovery from exercise and post-exertional exacerbation of symptoms are characteristic features of myalgic encephalomyelitis/chronic fatigue syndrome (ME/CFS). The gold standard to measure the degree of physical activity intolerance is cardiopulmonary exercise testing (CPET). Multiple studies have shown that peak oxygen consumption is reduced in the majority of ME/CFS patients, and that a 2-day CPET protocol further discriminates between ME/CFS patients and sedentary controls. Limited information is present on ME/CFS patients with a severe form of the disease. Therefore, the aim of this study was to compare the effects of a 2-day CPET protocol in female ME/CFS patients with a severe grade of the disease to mildly and moderately affected ME/CFS patients. Methods and results: We studied 82 female patients who had undergone a 2-day CPET protocol. Measures of oxygen consumption (VO_2_), heart rate (HR) and workload both at peak exercise and at the ventilatory threshold (VT) were collected. ME/CFS disease severity was graded according to the International Consensus Criteria. Thirty-one patients were clinically graded as having mild disease, 31 with moderate and 20 with severe disease. Baseline characteristics did not differ between the 3 groups. Within each severity group, all analyzed CPET parameters (peak VO_2_, VO_2_ at VT, peak workload and the workload at VT) decreased significantly from day-1 to day-2 (*p*-Value between 0.003 and <0.0001). The magnitude of the change in CPET parameters from day-1 to day-2 was similar between mild, moderate, and severe groups, except for the difference in peak workload between mild and severe patients (*p* = 0.019). The peak workload decreases from day-1 to day-2 was largest in the severe ME/CFS group (−19 (11) %). Conclusion: This relatively large 2-day CPET protocol study confirms previous findings of the reduction of various exercise variables in ME/CFS patients on day-2 testing. This is the first study to demonstrate that disease severity negatively influences exercise capacity in female ME/CFS patients. Finally, this study shows that the deterioration in peak workload from day-1 to day-2 is largest in the severe ME/CFS patient group.

## 1. Introduction

Myalgic Encephalomyelitis/Chronic Fatigue Syndrome (ME/CFS) is a serious and potentially disabling chronic disease [1,2,3,4]. As in other diseases, ME/CFS severity can range from mild to severe. Some patients can perform their daily activities at the expense of extra resting, while others are bed-ridden and dependent on others for help with activities of daily living. Exercise intolerance along with a prolonged recovery from activity (physical as well as mental) and post-exertional exacerbation of symptoms [4], represent an important characteristic of ME/CFS termed post-exertional malaise (PEM) [5,6].

Cardiopulmonary exercise testing (CPET) is the gold standard for measuring the degree of physical activity intolerance [7,8,9,10]. Multiple studies have shown that peak oxygen consumption is reduced in the majority of ME/CFS patients [11,12,13,14,15,16,17,18,19,20,21,22]. However, studies have also shown that a single CPET test in ME/CFS patients may show peak VO_2_ values that are similar to or only slightly lower than those of healthy sedentary controls [16]. To discriminate exercise capacity between ME/CFS patients and sedentary controls a 2-day CPET protocol has been proposed [19]. Studies using a 2-day CPET protocol, with two exercise tests separated by 24 h, have confirmed that ME/CFS patients have significantly lower VO_2_ and workload parameters on day 2 than on day 1. In contrast, sedentary controls have unaltered or slightly improved peak VO_2_ and workload [18,19,20,23,24]. We have recently confirmed the previous observations of a lower VO_2_ and workload in ME/CFS patients in a large group of male and female ME/CFS patients [25,26].

As peak oxygen consumption differs between males and females [27,28,29], the available studies were analyzed according to gender. Four studies reported peak oxygen consumption data in females only [18,19,20,25]. One study reported peak oxygen consumption in males [26] and two studies reported on combined information on males and females [15,24].

Limited studies have been published on ME/CFS patients with a severe form of the disease. The aim of this study was to compare 2-day CPET results from severely affected and mildly and moderately affected female ME/CFS patients using the severity grading as proposed by Carruthers et al., in the International Consensus Criteria [2]. We furthermore explored disability for both testing days with Weber’s disability metric [30].

## 2. Materials and Methods

Eligible participants were females with ME/CFS and exercise intolerance who had been referred to the Stichting Cardio Zorg, a cardiology clinic in the Netherlands that specializes in diagnosing and treating adults with ME/CFS. All patients underwent a detailed clinical history to establish the diagnosis of ME/CFS according to the ME criteria [2] and CFS criteria of Fukuda [1]. In all patients alternative diagnoses which could explain the fatigue and other symptoms were ruled out. The disease severity was scored according to the International Consensus Criteria. This was classified according to the paper as: “Symptom severity impact must result in a 50% or greater reduction in a patient’s premorbid activity level for a diagnosis of ME. Mild: approximately 50% reduction in activity, moderate: mostly housebound, severe: mostly bedbound and very severe: bedbound and dependent on help for physical functions” [2].

We reviewed the clinical records of 93 females with a diagnosis of ME/CFS who had undergone a 2-day CPET protocol between June 2012 and November 2019. Six patients were excluded because the ventilatory threshold could not be accurately determined. Two patients were excluded because of heart rate or blood pressure lowering drugs. Three patients were excluded because of being identified as being outliers. They were removed from further analysis. No patients were excluded due to insufficient effort during either day of testing, as judged by the supervising cardiologist. This left 82 female patients with available data from a 2-day CPET protocol for analysis.

All patients gave informed consent to analyze their data. The use of clinical data for descriptive studies was approved by the ethics committee of the Slotervaart Hospital, the Netherlands (reference number P1736).

### 2.1. Cardiopulmonary Exercise Testing (CPET)

Patients underwent a symptom-limited exercise test on a cycle ergometer (Excalibur, Lode, Groningen, The Netherlands) according to a previously described protocol [26]. Briefly, a ramp workload protocol was used varying between 10–30 Watt/min. Oxygen consumption (VO_2_), carbon dioxide release (VCO_2_), and oxygen saturation were continuously measured (Cortex, Procare, The Netherlands), and displayed on screen using Metasoft software (Cortex, Biophysic Gmbh, Germany). An ECG was continuously recorded and blood pressures were measured using the Nexfin device (BMEYE, Amsterdam, The Netherlands) [31]. The metabolic measurement system (Cortex, Biophysic Gmbh, Germany) was calibrated before each test with ambient air, standard gases of known concentrations, and a 3-L calibration syringe. The ventilatory threshold (VT), a measure of the anaerobic threshold, was identified from expired gases using the V-Slope algorithm [32]. The same experienced cardiologist supervised the 2 tests and performed visual assessment and confirmation of the algorithm-derived VT. The same cardiologists also ensured that the tests were done with the maximal effort possible for each specific patient. The mean of the VO_2_ measurements of the last 15 s before ending the exercise (peak VO_2_) was taken. VO_2_ at the peak and at the VT as well as the heart rate (HT) at peak exercise were expressed as a percentage of the normal values of a population study: %peak VO_2_, %VT VO_2_ [27]. Also, the mean respiratory exchange ratio (RER; VCO_2_/VO_2_) of the last 15 s was calculated by the software and presented in the results.

Maximum (or peak) oxygen uptake is used for the evaluation of cardiorespiratory endurance or aerobic fitness [7,8,33]. The anaerobic threshold (AT) was found to be an objective measure for aerobic work capacity [34,35,36]. Both invasive and non-invasive methods have been used for determining this value: invasive methods require blood lactate measurements (lactate threshold) and non-invasive methods rely on measuring respiratory gases, based on the relation between CO_2_ expiration and O_2_ inspiration (the respiratory exchange ratio or RER). Some studies have considered the AT to be at the point where the RER exceeds 1.0 [37,38,39]. As this has been considered inaccurate other methods have been proposed, like using the V-slope algorithm as described by Beaver et al. and used by others as in the current study [32]. In short, this method plots V_O2_ against V_CO2_. During aerobic metabolism the slope is slightly less than 1. With the onset of anaerobic metabolism, the slope increases to a value greater than 1, reflecting the production of extra CO_2_ resulting from HCO_3_^−^ buffering of lactate being produced. This point from slope 1 to the steeper slope 2 is called the ventilatory threshold (or lactic acidosis threshold) and is considered equivalent to the anaerobic threshold [32].

### 2.2. Disability Metric

In the early 1980s Weber et al. described a disability metric in evaluating heart failure patients with cardiopulmonary exercise testing [30]. This disability metric termed the classes A, B, C and D to avoid confusion with the New York Heart Association classification. The disability metric classifies how much impairment in aerobic capacity is present (Table 1).

### 2.3. Statistical Analysis

Data were analyzed using SPSS version 21 (IBM, Armonk, NY, USA). All continuous data were tested for normal distribution using the Shapiro-Wilk normality test, and presented as mean (SD) or as median (IQR), where appropriate. Nominal data (fibromyalgia and severity/disability) were compared using the chi-square test (a 3 × 2 table and a 4 × 3 table). For continuous data groups were compared using the paired or unpaired t-test where appropriate. Within group comparison was done by the ordinary one-way analysis of variance (ANOVA) or Kruskal-Wallis test where appropriate. Where significant, results were then explored further using the post-hoc Tukey’s test or Dunn’s test where appropriate. Within group comparison was done by the two-way analysis of variance (ANOVA). Where significant, results were then explored further using the post-hoc Holm-Sidak test. Graphpad Prism version 8.4.2 (Graphpad software, La Jolla, CA, USA) was used for the graphical representation of data in the figures.

## 3. Results

### 3.1. Baseline Characteristics

Table 2 shows the baseline characteristics of the 82 female ME/CFS patients. According to the ICC criteria, 31 were graded as mild, 31 were graded as moderate and 20 were graded as severe. No significant differences were found with respect to age, height, weight, BSA, BMI and disease duration. In the mild ME/CFS group 14 (45%) patients were classified as having comorbid fibromyalgia, in the moderate ME/CFS group 18 (58%) were classified as having fibromyalgia and in the severe ME/CFS group 12 (60%) were classified as having fibromyalgia (chi-square analysis 3 × 2 table: *p* = 0.48).

### 3.2. Two-Day CPET Data for ME/CFS Female Patients with Severe, Moderate and Mild Disease

Figure 1 shows the peak oxygen consumption for both CPET-1 and CPET-2 for mild, moderate and severe ME/CFS. For the mild disease group there was a significant decrease in peak VO_2_ of 2 mL/min/kg (−6%) (a change from 23 (5) to 21 (5) mL/min/kg; *p* = 0.003). For the moderate disease group there was a significant decrease in peak VO_2_ of 2 mL/min/kg (−11%) (a change from 17 (3) to 16 (4) mL/min/kg: *p* = 0.0001). For the severe disease group there was a significant decrease in peak VO_2_ of 2 mL/min/kg (−12%) (a change from 14 (3) to 12 (3) mL/min/kg: *p* = 0.003). Comparison of day 1 mild vs. moderate, mild vs. severe, and moderate vs. severe disease severity showed a significant difference between the groups (*p* ranging between 0.0001 and <0.0001). Comparison of day 2 mild vs. moderate, mild vs. severe, and moderate vs. severe disease severity showed a significant difference between the groups (all *p* < 0.0001).

Figure 2 shows the oxygen consumption at the ventilatory threshold for both CPET-1 and CPET-2 for mild, moderate and severe ME/CFS. For the mild disease group there was a significant decrease in VO_2_ at the ventilatory threshold of 3 mL/min/kg (−21%) (a change from 14 (2) to 11 (2) mL/min/kg; *p* < 0.0001). For the moderate disease group there was a significant decrease in VO_2_ at the ventilatory threshold of 3 mL/min/kg (−21%) (a change from 11 (2) to 9 (2) mL/min/kg: *p* < 0.0001). For the severe disease group there was a significant decrease in VO_2_ at the ventilatory threshold of 2 mL/min/kg (−19%) (a change from 10 (2) to 8 (2) mL/min/kg: *p* < 0.0001). Comparison of day 1 mild vs. moderate, mild vs. severe, and moderate vs. severe disease severity showed a significant difference between the groups (*p* ranging between 0.008 and <0.0001). Comparison of day 2 moderate vs. severe disease severity was not significantly different (*p* = 0.04). Comparing mild vs. moderate and mild vs. severe disease severity showed a significant difference between the groups (*p* ranging between 0.0007 and <0.0001).

Figure 3 shows the peak workload for both CPET-1 and CPET-2 for mild, moderate and severe ME/CFS. For the mild disease group there was a significant decrease in peak workload of 14 Watt (−10%) (a change from 144 (20) to 130 (22) Watt; *p* < 0.0001). For the moderate disease group there was a significant decrease in peak workload of 19 Watt (−16%) (a change from 117 (21) to 98 (24) Watt: *p* < 0.0001). For the severe disease group there was a significant decrease in peak workload of 17 Watt (−19%) (a change from 90 (26) to 73 (24) Watt: *p* < 0.0001). Comparison of day 1 mild vs. moderate, mild vs. severe, and moderate vs. severe disease severity showed a significant difference between the groups (*p* ranging between 0.0003 and <0.0001). Comparison of day 2 mild vs. moderate, mild vs. severe, and moderate vs. severe disease severity showed a significant difference between the groups (*p* ranging between 0.0003 and <0.0001).

Figure 4 shows the workload at the ventilatory threshold for both CPET-1 and CPET-2 for mild, moderate and severe ME/CFS. For the mild disease group there was a significant decrease in workload at the ventilatory threshold of 19 Watt (−26%) (a change from 69 (20) to 50 (19) Watt; *p* < 0.0001). For the moderate disease group there was a significant decrease in workload at the ventilatory threshold of 20 Watt (−31%) (a change from 61 (19) to 41 (15) Watt: *p* < 0.0001). For the severe disease group there was a significant decrease in workload at the ventilatory threshold of 18 Watt (−33%) (a change from 53 (19) to 36 (16) Watt: *p* < 0.0001). Comparison of day 1 mild vs. moderate and moderate vs. severe disease severity were not significantly different (*p* = 0.15 and 0.13 respectively. Comparison between mild vs. severe disease severity was significantly different (*p* = 0.006). Comparison of day 2 mild vs. moderate and moderate vs. severe disease severity were not significantly different (*p* = 0.21 and 0.05 respectively. Comparison between mild vs. severe disease severity was significantly different (*p* = 0.006).

### 3.3. Comparison of ME/CFS Patients with Severe, Moderate and Mild Disease for CPET Day-1 and Day-2 Variables

Table 3 shows the percent difference in CPET parameters from CPET-2 and CPET-1 for VO_2_ VT, VO_2_ peak, heart rate at the VT and peak exercise, and workload at the VT and at peak exercise for severe, moderate and mild ME/CFS patients. The post-hoc analysis showed that there was only a significantly higher decrease in the percent change in peak workload of severe patients compared to mild patients (*p* = 0.019).

Figure 5 shows the subdivision of mild, moderate and severe disease severity and Weber disability grades A-D for both CPET-1 and CPET-2 (panel A: mild ME/CFS CPET-1; panel B: mild ME/CFS CPET-2; panel C: moderate ME/CFS CPET-1; panel D: moderate ME/CFS CPET-2; panel E: severe ME/CFS CPET-1; panel F: severe ME/CFS CPET-2). A clear shift is visible for all three severity groups with more disability on day-2. Chi square testing was highly significantly different for both day-1 and day-2 between the three severity groups (*p* < 0.0001).

Table 4 shows the RER results for each severity group by CPET day. A two-way analysis showed no significance for interaction between the 3 disease severity groups and either CPET day (*p* = 0.59).

## 4. Discussion

The main finding of this study is that with a 2-day CPET protocol there is a consistent decrease from mildly affected to severely affected ME/CFS patients in peak oxygen consumption, oxygen consumption at the ventilatory threshold, peak workload and workload at the ventilatory threshold. We demonstrated a greater degree of disability in all disease severities when CPET-1 was compared to CPET-2. We believe this is the first study of 2-day CPET protocols in ME/CFS patients to include severity grading in the analysis [15,18,19,20,23,24,25,26].

### 4.1. Two-Day Cardiopulmonary Exercise Test Studies Reported in Literature

Eight studies have reported the results of 2-day CPETs in male and/or female ME/CFS patients. The relevant CPET parameters for female ME/CFS patients are presented in Table 5. All studies reported peak oxygen consumption and oxygen consumption at the ventilatory threshold [15,18,19,20,23,24,25,26]. Exact data could not be derived from one study, which was excluded in this overview [23]. Three studies reported percent peak oxygen consumption [15,25,26]. Two studies reported the percent oxygen consumption at the ventilatory threshold [25,26]. Six studies reported both the peak workload and workload at the ventilatory threshold [15,18,19,20,24,25,26].

Our lowest values were measured in severe ME/CFS patients, which are lower than results reported in the literature. It is therefore less likely that the more severely affected patients were included in these previous studies. Further studies of exercise intolerance from severely affected ME/CFS patients should improve our understandings of the true spectrum of disease severity.

Interestingly, with increasing severity, the percentage decrease in peak oxygen consumption from day-1 to day-2 was not significantly different between mild, moderate and severe patients, whereas the percentage decrease in peak workload from day-1 to day-2 was significantly different only between mild and severe patients. The larger percentage decrease in workload at day-2 in severe ME/CFS patients compared to mild and moderate ME/CFS patients, may indicate a more pronounced absence of recovery from day-1 as a measure of post-exertional malaise. This also needs to be confirmed in larger prospective studies.

The change in Weber’s disability metric is an alternative validation of the clinical severity grading from ICC. The increase in disability grading on day-2 versus day-1 supports the severity classification. The concordance between the Weber disability metric and the ICC severity grading was not examined as part of this study.

We included patients who reached a maximal clinical effort as judged by the supervising cardiologist, even if the RER was below 1.1. Although an RER of less than 1.1 is often viewed as indicating inadequate effort, patients with severely impaired exercise tolerance can develop skeletal muscle exhaustion earlier than central hemodynamic and ventilatory factors become limiting. This in turn interrupts exercise at peak respiratory exchange ratio values lower than 1.00 [40]. Metabolic skeletal muscle abnormalities are present ME/CFS [6,41,42], and pain in those with co-morbid fibromyalgia can also limit maximal exercise performance. In support of including those with maximal clinical effort but an RER less than 1.1, we have recently shown that adults with ME/CFS with and without an RER of at least 1.1 did not differ with regard to correlations between other measures of exercise, including the physical activity subscale of the SF-36, peak oxygen consumption and the number of steps per day [43]. In those with a concomitant diagnosis of fibromyalgia the RER in that study was significantly lower than in the ME/CFS patients without fibromyalgia; exercise in this subgroup was terminated due to of muscle pain. The %peak VO_2_, the number of steps and the physical functioning scale were not different between ME/CFS patients with and without fibromyalgia. The results of the current study are consistent with the earlier report, and argue for including in ME/CFS studies those who meet criteria for maximal effort as judged by a clinician even when the RER is less than 1.1. Excluding these patients has the potential to underestimate the severity of the activity limitations in ME/CFS.

### 4.2. Limitations

First, we did not include a group of sedentary controls for comparison. Second, this was not a prospective trial, as patients underwent the 2-day CPET protocol not only for clinical testing, but also for security claim reasons or to examine the hypothesis that deconditioning accounted for the results. Poor exercise results during testing and in daily life are often thought to be caused by deconditioning rather than being the result of a disease with prominent disabilities. This may have led to inclusion bias.

## 5. Conclusions

This large 2-day CPET protocol study confirms and extends previous findings of the reduction of various exercise variables in ME/CFS patients on day-2. This is the first study to demonstrate that disease severity negatively influences exercise capacity in female ME/CFS patients, confirming that deterioration in peak workload from day-1 to day-2 is largest in the severe ME/CFS patient group.

## Figures and Tables

**Figure 1 healthcare-08-00192-f001:**
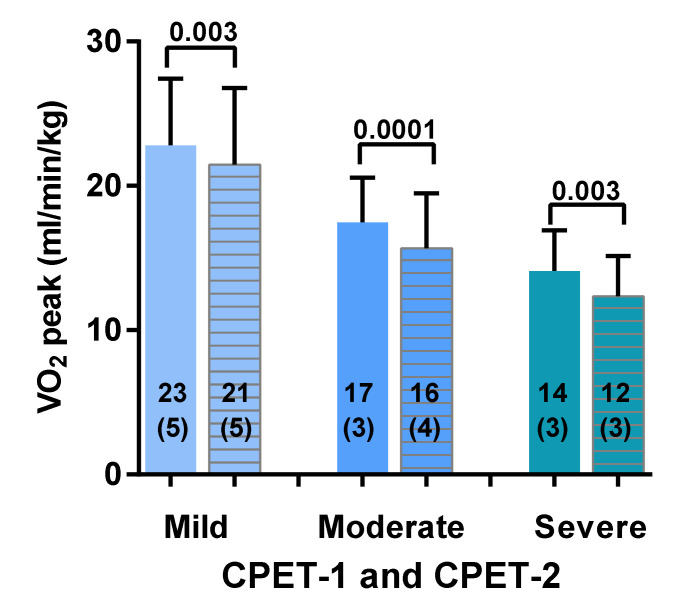
Peak oxygen consumption for both CPET-1 and CPET-2 for mild, moderate and severe ME/CFS. CPET: cardiopulmonary exercise test; VO_2_: oxygen consumption. For between group comparisons see the result section. CPET-1 is presented by a solid bar and CPET-2 is represented by a lined bar.

**Figure 2 healthcare-08-00192-f002:**
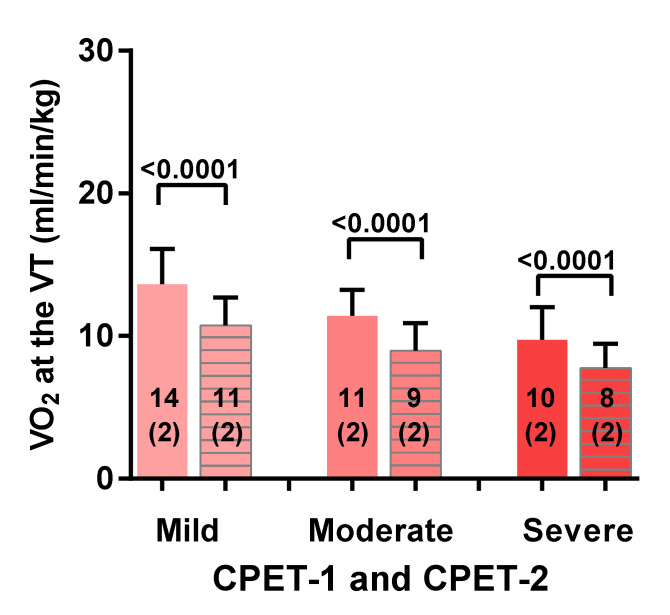
Oxygen consumption at the ventilatory threshold for both CPET-1 and CPET-2 for mild, moderate and severe ME/CFS patients. CPET: cardiopulmonary exercise test; VO_2_: oxygen consumption; VT: ventilatory (or anaerobic) threshold. For between group comparisons see the result section. CPET-1 is presented by a solid bar and CPET-2 is represented by a lined bar.

**Figure 3 healthcare-08-00192-f003:**
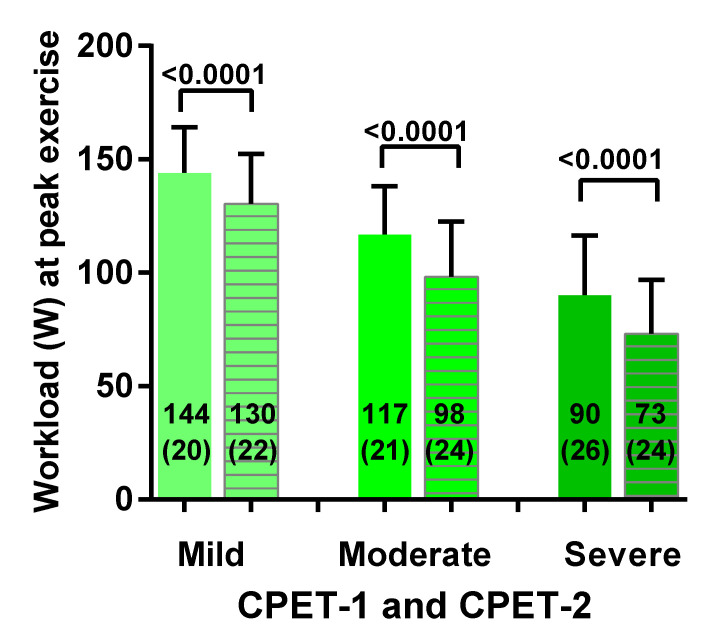
Peak workload for both CPET-1 and CPET-2 for mild, moderate and severe ME/CFS patients. CPET: cardiopulmonary exercise test. For between group comparisons see the result section. CPET-1 is presented by a solid bar and CPET-2 is represented by a lined bar.

**Figure 4 healthcare-08-00192-f004:**
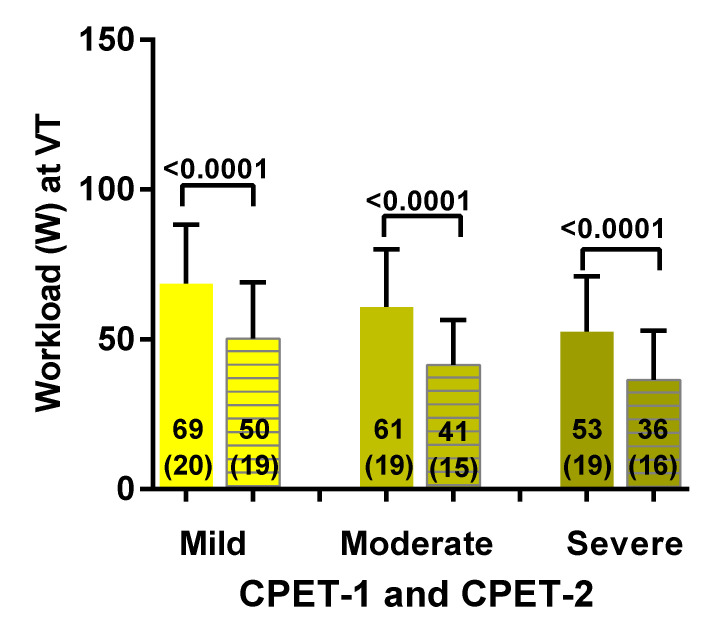
Workload at the ventilatory threshold for both CPET-1 and CPET-2 for mild, moderate and severe ME/CFS patients. CPET: cardiopulmonary exercise test; VT: ventilatory (or anaerobic) threshold. For between group comparisons see the result section. CPET-1 is presented by a solid bar and CPET-2 is represented by a lined bar.

**Figure 5 healthcare-08-00192-f005:**
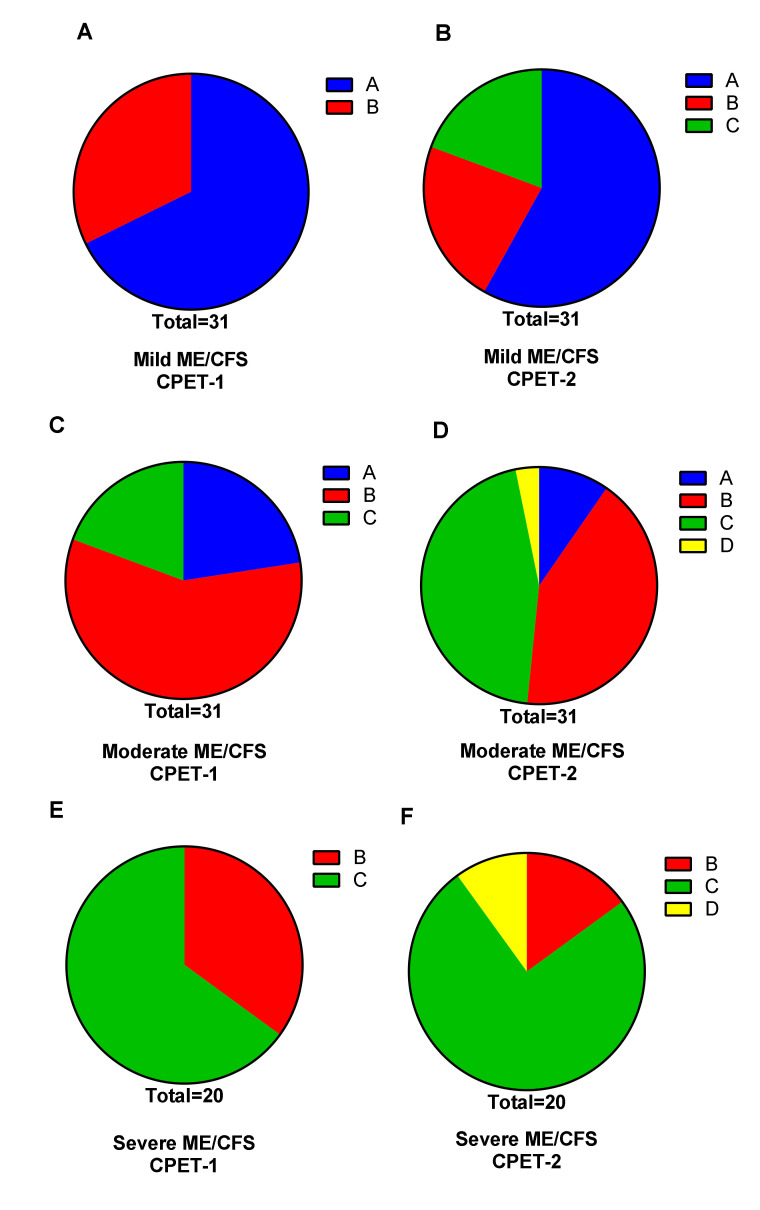
Disability grading according to Weber on both CPET-1 and CPET-2 for mild, moderate and severe ME/CFS patients. CPET: cardiopulmonary exercise test; Weber grading (**A**) (>20 mL/min/kg) (blue), (**B**) (16–20 mL/min/kg) (red), (**C**) (10–15 mL/min/kg) (green) and (**D**) (<10 mL/min/kg) (yellow). Panels (**A**,**C**,**E**) are representing CPET-1 for respectively mild, moderate and severe disease. Panels (**B**,**D**,**F**) are representing CPET-2 for respectively mild, moderate and severe disease [30].

**Table 1 healthcare-08-00192-t001:** Weber disability metric (30) [30].

Weber Class	Oxygen Consumption	Aerobic Capacity
Class A	>20 mL/min/kg	No impairment
Class B	16–20 mL/min/kg	Mild to moderate impairment
Class C	10–15 mL/min/kg	Moderate to severe impairment
Class D	<10 mL/min/kg	Severe impairment

**Table 2 healthcare-08-00192-t002:** Baseline criteria for female ME/CFS patients.

	Group 1 Severe (*n* = 20)	Group 2 Moderate (*n* = 31)	Group 3 Mild (*n* = 31)	ANOVA/Kruskal-Wallis Test
Age (years)	39 (10)	41 (10)	42 (9)	F (2, 79) = 0.46; *p* = 0.63
Height (cm)	171 (7)	171 (7)	169 (6)	F (2, 79) = 1.56; *p* = 0.22
Weight (kg)	70 (61–77)	69 (63–80)	65 (60–72)	X^2^(2) = 1.144; *p* = 0.56
BSA (m^2^)	1.4 (1.2–1.8)	1.4 (1.3–1.5)	1.3 (1.2–1.4)	X^2^(2) = 1.690; *p* = 0.43
BMI (kg/m^2^)	22.7 (21.8–27.6)	23.9 (20.7–27.6)	23.4 (21.4–26.7)	X^2^(2) = 0.032; *p* = 0.98
Disease duration (years)	15.9 (9.3)	13.3 (8.9)	13.5 (9.3)	F (2, 79) = 0.56; *p* = 0.57

Data are presented as mean (SD) or median (IQR). BMI: body mass index (DuBois formula); BSA: body surface area.

**Table 3 healthcare-08-00192-t003:** Percent differences from CPET-2 minus CPET-1 comparing female ME/CFS patients with severe, moderate and mild disease severity.

Percent Difference CPET-2 Minus CPET-1	Group 1 Severe (*n* = 20)	Group 2 Moderate (*n* = 31)	Group 3 Mild (*n* = 31)	ANOVA and Post-Hoc Tukey’s Test
VO_2_ peak (mL/min/kg)	−12 (14)	−11 (14)	−6 (11)	F (2, 79) = 1.28; *p* = 0.28
HR peak (bpm)	−7 (6)	−6 (7)	−3 (6)	F (2, 79) = 2.08; *p* = 0.13
Workload peak (Watts)	−19 (11)	−16 (15)	−10 (8)	F (2, 79) = 4.37; *p* = 0.016. Post-hoc tests: 1 vs. 2 *p* = 0.083; 1 vs. 3 *p* = 0.019 and 2 vs. 3 *p* = 0.68
VO_2_ VT (mL/min/kg)	−19 (11)	−21 (12)	−21 (11)	F (2, 79) = 0.26; *p* = 0.77
HR VT (bpm)	−7 (5)	−9 (6)	−8 (7)	F (2, 79) = 0.78; *p* = 0.46
Workload VT (Watts)	−33 (20)	−31 (18)	−26 (18)	F (2, 79) = 0.54; *p* = 0.58
RER	−5 (7)	−2 (9)	−3 (7)	F (2, 79) = 0.83; *p* = 0.44

Data are presented as mean (SD). VT: ventilatory threshold; CPET: cardiopulmonary exercise test; HR: heart rate; VO_2_: oxygen consumption; RER: respiratory exchange ratio.

**Table 4 healthcare-08-00192-t004:** RER values for each severity group on each day.

Group 1 Severe (*n* = 20)	Group 2 Moderate (*n* = 31)	Group 3 Mild (*n* = 31)	ANOVA
CPET day-1			
1.08 (0.09)	1.09 (0.09)	1.13 (0.11)	F(2.79) = 2.02; *p* = 0.14
CPET day-2			
1.02 (0.11)	1.07 (0.11)	1.10 (0.11	F(2.79) = 2.77; *p* = 0.07

**Table 5 healthcare-08-00192-t005:** Ranges of CPET parameters (oxygen consumption at peak exercise and at the ventilatory threshold, workload at peak exercise and at the ventilatory threshold) in previous literature ranging from the minimal to the maximal value reported on day-1 and day-2 and the ranges from the CPET values from the present study ranging from the minimum value (severe disease) to the maximal value (mild disease) [15,18,19,20,24,25].

	Literature Day-1	Literature Day-2	Present Study Day-1	Present Study Day-2
Peak VO_2_ (mL/min/kg)	19–26	17–21	14–23	12–21
VT VO_2_ (mL/min/kg)	12–15	9–12	10–14	8–11
Peak Workload (Watt)	110–132	102–125	90–144	73–130
VT Workload (Watt)	50–62	22–54	53–69	36–50

VO_2_: oxygen consumption; VT: ventilatory threshold.

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
