# Peer review of "Two-Day Cardiopulmonary Exercise Testing in Females with a Severe Grade of Myalgic Encephalomyelitis/Chronic Fatigue Syndrome: Comparison with Patients with Mild and Moderate Disease"

_healthcare, 2020, doi:10.3390/healthcare8030192_

Round 1

Reviewer 1 Report

Overall, a sound and clearly presented study with relevant results.  Applicability of results in the Discussion section could be enhanced by adding a brief section on the difference in a commonly-used disability metric (e.g. Weber & Janicki or AMA guidelines) to illustrate the magnitude of change due to PEM in terms of disability ranking from test 1 to test 2 in each of the illness severity groups.  I don't consider this to be essential for publication, but since most or all of the 2-day CPETs were conducted to provide evidence of impairment for social security, it would be useful to show if or how much decrement in an impairment rating a patient experiences due to PEM, and if that decrement in rating is most impactful in the mild, moderate or severe patient.  For example, does the mild ME patient drop to an impairment rating of 'moderate' due to PEM (based on test 2 results)?

Errors/edits that need attention:

Spacing errors on lines 23,26,46,50,52,53,56,68,85,90,93,98,136 (Range of absolute differences... should be "-6 to 4), 154,201,208,210,212,213

Errors in reference list include errors in spacing of journal names on lines 259,261,267,270,272,274,279,284,286,288,293,297,

Errors in spelling in references include lines 289 (Stevens should be capitalized), 304 - should Van Campen be the same on lines 304 and 306?

Edits on line 59 (change "in ME/CFS" to "on ME/CFS"; line 77 (remove "be" after "accurately"); line 85 (change "RAMP" to "ramp" - 'ramp' is not an acronym and should not be in caps); line 56 - ref. 11 study did not include control subjects; line 96 - delete "the" after "(HT) at the peak exercise"

STATS - line 104 - it's unclear how Chi-squared was applied to "gender" since all subjects were females.  Should this be Chi-squared applied to "illness severity group"?

RESULTS - There are some errors in data presentation especially pertaining to figures.  Assuming that data in Tables 2, a, b & c are correct, then some data in the figures and figure legends do not correspond to data in the tables.  

Specifically, line 146 - change 22 (5) to 23 (5) to be consistent with data in Table 2c.  

Also line 165 - change 140 to 145 and 132 to 130

line 166 - change 70 (31) to 75 (29)

line 174 - change 69 to 68 and 51 to 50

The values also appear in the corresponding bar graphs and should be corrected as well.

Author Response

Bovenkant formulier

Open Review

English language and style

( ) Extensive editing of English language and style required
( ) Moderate English changes required
(x) English language and style are fine/minor spell check required
( ) I don't feel qualified to judge about the English language and style

Yes

Can be improved

Must be improved

Not applicable

Does the introduction provide sufficient background and include all relevant references?

(x)

( )

( )

( )

Is the research design appropriate?

(x)

( )

( )

( )

Are the methods adequately described?

(x)

( )

( )

( )

Are the results clearly presented?

( )

( )

(x)

( )

Are the conclusions supported by the results?

(x)

( )

( )

( )

Comments and Suggestions for Authors

Overall, a sound and clearly presented study with relevant results.  Applicability of results in the Discussion section could be enhanced by adding a brief section on the difference in a commonly-used disability metric (e.g. Weber & Janicki or AMA guidelines) to illustrate the magnitude of change due to PEM in terms of disability ranking from test 1 to test 2 in each of the illness severity groups.  I don't consider this to be essential for publication, but since most or all of the 2-day CPETs were conducted to provide evidence of impairment for social security, it would be useful to show if or how much decrement in an impairment rating a patient experiences due to PEM, and if that decrement in rating is most impactful in the mild, moderate or severe patient.  For example, does the mild ME patient drop to an impairment rating of 'moderate' due to PEM (based on test 2 results)?

We applied the Weber criteria on the population and added this information to the methods, results and discussion section.

Errors/edits that need attention:

Spacing errors on lines 23,26,46,50,52,53,56,68,85,90,93,98,136 (Range of absolute differences... should be "-6 to 4), 154,201,208,210,212,213 checked and adapted

Errors in reference list include errors in spacing of journal names on lines 259,261,267,270,272,274,279,284,286,288,293,297,checked and adapted

Errors in spelling in references include lines 289 (Stevens should be capitalized), 304 - should Van Campen be the same on lines 304 and 306? Checked and adapted

Edits on line 59 (change "in ME/CFS" to "on ME/CFS"; line 77 (remove "be" after "accurately"); line 85 (change "RAMP" to "ramp" - 'ramp' is not an acronym and should not be in caps); line 56 - ref. 11 study did not include control subjects; line 96 - delete "the" after "(HT) at the peak exercise" altered

STATS - line 104 - it's unclear how Chi-squared was applied to "gender" since all subjects were females.  Should this be Chi-squared applied to "illness severity group"? updated

RESULTS - There are some errors in data presentation especially pertaining to figures.  Assuming that data in Tables 2, a, b & c are correct, then some data in the figures and figure legends do not correspond to data in the tables.  

Specifically, line 146 - change 22 (5) to 23 (5) to be consistent with data in Table 2c.  

Also line 165 - change 140 to 145 and 132 to 130

line 166 - change 70 (31) to 75 (29)

line 174 - change 69 to 68 and 51 to 50

The values also appear in the corresponding bar graphs and should be corrected as well. Checked and corrected as appropriate for text, tables and figures.

Submission Date

28 April 2020

Date of this review

08 May 2020 16:48:32

Reviewer 2 Report

This is an excellent, very well written paper looking at cardiopulmonary exercise testing for patients with CFS/ME

I have only minor comments:

1) To adjust for the multiple analyses did the authors consider using a Bonferroni correction rather than the somewhat arbitrary threshold of p<0.01. This is unlikely to change the conclusions drawn, but would give the authors a more credible threshold to base their decisions on. 

2) The first two sentences in the results section I think are editorial notes and should be removed.

3) In 4.2 Limitations, I wasn't clear what was meant by "or to rebuttal a hypothesis if deconditioning". Is there a way to explain this better?

Author Response

Open Review

English language and style

( ) Extensive editing of English language and style required
( ) Moderate English changes required
(x) English language and style are fine/minor spell check required
( ) I don't feel qualified to judge about the English language and style

Yes

Can be improved

Must be improved

Not applicable

Does the introduction provide sufficient background and include all relevant references?

(x)

( )

( )

( )

Is the research design appropriate?

(x)

( )

( )

( )

Are the methods adequately described?

(x)

( )

( )

( )

Are the results clearly presented?

(x)

( )

( )

( )

Are the conclusions supported by the results?

(x)

( )

( )

( )

Comments and Suggestions for Authors

This is an excellent, very well written paper looking at cardiopulmonary exercise testing for patients with CFS/ME

I have only minor comments:

1) To adjust for the multiple analyses did the authors consider using a Bonferroni correction rather than the somewhat arbitrary threshold of p<0.01. This is unlikely to change the conclusions drawn, but would give the authors a more credible threshold to base their decisions on. To the one way ANOVA analysis, the post-hoc analysis according to Tukey’s analysis of multiple comparisons was added. The results were simplified for clarity purposes. As was where appropriate the Kuskal-Wallis with post-hoc’s Dunn correction.

2) The first two sentences in the results section I think are editorial notes and should be removed. You’re so right. Thank you so much.

3) In 4.2 Limitations, I wasn't clear what was meant by "or to rebuttal a hypothesis if deconditioning". Is there a way to explain this better? We updated to clarify.

Submission Date

28 April 2020

Date of this review

15 May 2020 14:15:26

Reviewer 3 Report

42 -44 As in other diseases, ME/CFS severity can range from mild, in which patients can perform their daily activities at the expense of extra resting, to very severe, in which patients are bed-ridden and are dependent on others for help with activities of daily living. (reads as if ME/CFS symptoms also apply to other diseases)

Suggest: As in other diseases, ME/CFS severity can range from mild to severe. Some patients can perform their daily activities at the expense of extra resting, while others  are bed-ridden and  dependent on others for help with activities of daily living.

44-47 An  important symptom of ME/CFS patients is exercise intolerance along with a prolonged recovery from 46 exercise (physical as well as mental) and post-exertional exacerbation of ME/CFS symptoms(4), 47 termed post-exertional malaise (PEM)(5, 6). (passive sentence)

Suggest: Exercise intolerance along with a prolonged recovery from activity (physical as well as mental) and post-exertional exacerbation of  symptoms(4) represent an important characteristic of ME/CFS termed  post-exertional malaise (PEM)(5, 6).

48 The gold standard for measuring the degree of physical activity intolerance is cardiopulmonary 49 exercise testing (CPET).

Suggest: Cardiopulmonary  exercise testing (CPET) is the gold standard for measuring degree of physical activity intolerance. (inc. ref.)

59 Limited studies have been published in ME/CFS patients with a severe form of the disease. The 60 aim of this study was to report the results of a 2-day CPET protocol in female ME/CFS patients with 61 a severe grade of disease. CPET data of severely affected female ME/CFS patients were compared 62 with the results of mildly and moderately affected patients, using the severity grading as proposed 63 by Carruthers et al., in the International Consensus Criteria(2). (repetition,

Suggest: Limited studies have been published in ME/CFS patients with a severe form of the disease. The aim of this study was to compare 2-day CPET results from severely affected and mildly and moderately affected female ME/CFS patients using the severity grading as proposed by Carruthers et al., in the International Consensus Criteria(2).

83 Explain key measures VO2 peak, V/AT, RER, VCO2/VO2 and why important. Likely that the audience will include other than exercise physiologists

92 Explain relationship between VT and AT.

105 of the three groups. Because of the multiple comparisons a conservative p value of <0.01 was 106 considered significant. Bonferoni? Analysis is very complex. Would a more parsimonious approach work? Why didn’t you use Group(3) x Test ANOVA’s for each ind. Var. or even 3 x 2  MANOVA for all ind. vars ?)

108 This section may be divided by subheadings. It should provide a concise and precise description 109 of the experimental results, their interpretation as well as the experimental conclusions that can be 110 drawn. (? Delete).

Figure 1 shows the peak oxygen consumption for both CPET-1 and CPET-2 for mild, moderate 144 and severe ME/CFS. All values between day 1 and day 2 differed significantly for all three groups 145 (p<0.0001). Peak oxygen consumption changed from 22 (5) to 21 (5) ml/min/kg in mild disease, from 146 18 (3) to 16 (4) ml/min/kg in moderate disease and 14 (3) to 12 (3) ml/min/kg in severe disease (all 147 p<0.0001). ANOVA showed a highly significant difference between the groups (p<0.0001).

Suggest: For the  mild disease group there was a significant decrease in VO2 peak of X ml/kg/min  (X%) between tests (mean =X, sd=X vs mean=X, sd=X, p=X). REPEAT FOR EACH GROUP

ANOVA showed a highly significant difference between the groups (p<0.0001). ANOVA comparisons not clear. Test 1 vs Test 2 for each group? T1: g1 vs g2 vs g3; T2 g1 vs g2 vs g3? Post hoc analyses for multiple group ANOVAS?.

(116) The tests 116 were performed for social security claims and for rebuttal of the deconditioning hypothesis raised by 117 other physicians. (I think this should be with patient eligibility line 65).

Results are difficult to read. Suggest simplifying where possible. State  significant results first followed by non-significant results. Tables: units in parentheses e.g., (bpm), will simplify tables. Need to state somewhere where data represents mean and (SD).

As peak oxygen consumption differs between males and females(25-27), the available studies 211 were analyzed according to gender. Four studies reported peak oxygen consumption data in females 212 only(14-16, 22). One study reported peak oxygen consumption in males(21) and two studies reported 213 on combined information on males and females(11, 20). (This really belongs in the Intro. Use to justify why only females in this current study).

Discussion: You are comparing group means among and within studies from the literature? Really needs a better summary of your findings. What do you think were the most important findings? Not really much discussion of functional/clinical implications of your findings. Can you expand on the significance of the observed differences T1 to T2 among groups? Can you comment on value of 2-day CPET in documenting PEM, i.e., what did you observe with repeated CPET that would not have been apparent from a single CPET? Can you comment further on possible mechanism(s) underlying the results?

Overall, paper would benefit from further editing. Suggest reviewing for passive sentences and making active where possible to reduce complexity and improve readability. Simplify results section to make it easier to read. Be sure to provide “the experimental conclusions”.

Author Response

Open Review

English language and style

( ) Extensive editing of English language and style required
(x) Moderate English changes required
( ) English language and style are fine/minor spell check required
( ) I don't feel qualified to judge about the English language and style

Yes

Can be improved

Must be improved

Not applicable

Does the introduction provide sufficient background and include all relevant references?

( )

(x)

( )

( )

Is the research design appropriate?

(x)

( )

( )

( )

Are the methods adequately described?

( )

(x)

( )

( )

Are the results clearly presented?

( )

( )

(x)

( )

Are the conclusions supported by the results?

( )

(x)

( )

( )

Comments and Suggestions for Authors

42 -44 As in other diseases, ME/CFS severity can range from mild, in which patients can perform their daily activities at the expense of extra resting, to very severe, in which patients are bed-ridden and are dependent on others for help with activities of daily living. (reads as if ME/CFS symptoms also apply to other diseases)

Suggest: As in other diseases, ME/CFS severity can range from mild to severe. Some patients can perform their daily activities at the expense of extra resting, while others  are bed-ridden and  dependent on others for help with activities of daily living. Thank you for the suggestion: adapted.

44-47 An  important symptom of ME/CFS patients is exercise intolerance along with a prolonged recovery from 46 exercise (physical as well as mental) and post-exertional exacerbation of ME/CFS symptoms(4), 47 termed post-exertional malaise (PEM)(5, 6). (passive sentence)

Suggest: Exercise intolerance along with a prolonged recovery from activity (physical as well as mental) and post-exertional exacerbation of symptoms(4) represent an important characteristic of ME/CFS termed post-exertional malaise (PEM)(5, 6). Thank you for the suggestion: adapted.

48 The gold standard for measuring the degree of physical activity intolerance is cardiopulmonary 49 exercise testing (CPET).

Suggest: Cardiopulmonary exercise testing (CPET) is the gold standard for measuring degree of physical activity intolerance. (inc. ref.) Thank you for the suggestion: adapted.

59 Limited studies have been published in ME/CFS patients with a severe form of the disease. The 60 aim of this study was to report the results of a 2-day CPET protocol in female ME/CFS patients with 61 a severe grade of disease. CPET data of severely affected female ME/CFS patients were compared 62 with the results of mildly and moderately affected patients, using the severity grading as proposed 63 by Carruthers et al., in the International Consensus Criteria(2). (repetition,

Suggest: Limited studies have been published on ME/CFS patients with a severe form of the disease. The aim of this study was to compare 2-day CPET results from severely affected and mildly and moderately affected female ME/CFS patients using the severity grading as proposed by Carruthers et al., in the International Consensus Criteria(2). Thank you for the suggestion: adapted.

83 Explain key measures VO2 peak, V/AT, RER, VCO2/VO2 and why important. Likely that the audience will include other than exercise physiologists added some additional information, especially for readers not familiar with CPET parameters.

92 Explain relationship between VT and AT. added

105 of the three groups. Because of the multiple comparisons a conservative p value of <0.01 was 106 considered significant. Bonferoni? Analysis is very complex. Would a more parsimonious approach work? Why didn’t you use Group(3) x Test ANOVA’s for each ind. Var. or even 3 x 2  MANOVA for all ind. vars ?). We used another more extended statistical software package to re-analyze and explore the suggestions. We thank the reviewer for his suggestions, it improved the paper.The analysis was simplified to show the results more clearly. On the tables with 3 groups the one way ANOVA analysis with post-hoc Tukey’s analysis for multiple comparisons or the Kuskal-Wallis with post hoc Dunn correction where appropriate was used. Graphpad prism was used for graphical representation.

108 This section may be divided by subheadings. It should provide a concise and precise description 109 of the experimental results, their interpretation as well as the experimental conclusions that can be 110 drawn. (? Delete). deleted

Figure 1 shows the peak oxygen consumption for both CPET-1 and CPET-2 for mild, moderate 144 and severe ME/CFS. All values between day 1 and day 2 differed significantly for all three groups 145 (p<0.0001). Peak oxygen consumption changed from 22 (5) to 21 (5) ml/min/kg in mild disease, from 146 18 (3) to 16 (4) ml/min/kg in moderate disease and 14 (3) to 12 (3) ml/min/kg in severe disease (all 147 p<0.0001). ANOVA showed a highly significant difference between the groups (p<0.0001). ANOVA removed from figures to simplify. Between group comparisons described in result section.

Suggest: For the  mild disease group there was a significant decrease in VO2 peak of X ml/kg/min  (X%) between tests (mean =X, sd=X vs mean=X, sd=X, p=X). REPEAT FOR EACH GROUP Adapted (in a similar fashion also for figures 2-4).

ANOVA showed a highly significant difference between the groups (p<0.0001). ANOVA comparisons not clear. Test 1 vs Test 2 for each group? T1: g1 vs g2 vs g3; T2 g1 vs g2 vs g3? Post hoc analyses for multiple group ANOVAS?. Added post-hoc analysis according to Tukey’s test for multiple comparisons (or Kuskal-Wallis with Dunn where appropriate). To clarify figures comparisons between days were mentioned in the text not to complicate the figures.

(116) The tests 116 were performed for social security claims and for rebuttal of the deconditioning hypothesis raised by 117 other physicians. (I think this should be with patient eligibility line 65). Clinical testing was the primary reason for testing. Social security claims and rebuttal of deconditioning was secondary reasons for performing the test.

Results are difficult to read. Suggest simplifying where possible. State  significant results first followed by non-significant results. Tables: units in parentheses e.g., (bpm), will simplify tables. Need to state somewhere where data represents mean and (SD). Adapted

As peak oxygen consumption differs between males and females(25-27), the available studies 211 were analyzed according to gender. Four studies reported peak oxygen consumption data in females 212 only(14-16, 22). One study reported peak oxygen consumption in males(21) and two studies reported 213 on combined information on males and females(11, 20). (This really belongs in the Intro. Use to justify why only females in this current study). As suggested shifted to introduction.

Discussion: You are comparing group means among and within studies from the literature? Really needs a better summary of your findings. What do you think were the most important findings? Not really much discussion of functional/clinical implications of your findings. Can you expand on the significance of the observed differences T1 to T2 among groups? Can you comment on value of 2-day CPET in documenting PEM, i.e., what did you observe with repeated CPET that would not have been apparent from a single CPET? Can you comment further on possible mechanism(s) underlying the results? Clarified by presenting literature results and current results in a table.

Overall, paper would benefit from further editing. Suggest reviewing for passive sentences and making active where possible to reduce complexity and improve readability. Simplify results section to make it easier to read. Be sure to provide “the experimental conclusions”.

Round 2

Reviewer 3 Report

109        the AT to be at the point where the RER exceeds 1.0 (37, 38), but because considered inaccuracy other  Does not make sense

115 ventilatory threshold (or lactate acidosis threshold) and is considered equivalent to the anaerobic

116 threshold. Add reference

119 patients with cardiopulmonary exercise testing. This disability metric was described as: “We termed

120 the classes A, B, C and D to avoid confusion with the New York Heart Association classification.

121 Functional class A represents little or no impairment in aerobic capacity and is present when VO2 max

122 exceeds 20 ml/min/kg. For age- and sex-corrected VO2 max of the normal adult population commonly

123 seen in practice, VO2 max will be above this level. Class B represents mild-to-moderate impairment

124 and is present when VO2 max ranges from 16 to 20 ml/min/kg. Class C represents moderate-to-severe

125 impairment, and is present when VO2 max falls between 10 and 15 ml/min/kg. Class D represents a

126 severe impairment, with a VO2 max of <10 ml/min/kg” (30). Overlong for a direct quote and grammar does fit with rest of paragraph. Suggest paraphrasing or include as a figure

143 In the mild ME/CFS group 14 (45%) patients were classified as having fibromyalgia, in the moderate

144 ME/CFS group 18 (58%) were classified as having fibromyalgia and in the severe ME/CFS group 12

145 (60%) were classified as having fibromyalgia (chi-square analysis 3x2 table: p=0.48). Add “comorbid” before Fibromyalgia”. Could also mention incidence of comorbid ME/CFS and FMS with ref.

195 severe, moderate and mild ME/CFS patients. The post-hoc analysis showed that there was only a

196 significantly higher decrease in the percent change in peak workload of severe patients compared to

197 mild patients (p=0.019).

Suggest: The post-hoc analysis showed a significantly higher decrease in the percent change in peak workload of severe patients compared to mild patients (p=0.019). No other comparisons were significant (p values?).

Tables. Suggest

Group                  1. Severe (n=20)              2. Moderate (n=31)        3. Mild (n=31)

This will lose a line and  simplify tables

Add RER to Table 2.

Overall, paper is clearer and much easier to read. I am, however, concerned that RER data is not reported or included in the analyses. As the best measure of patient effort, knowledge of peak RER is essential to the interpretation of group and test differences. This is an important omission and needs to be corrected before publication.

Round 3

Reviewer 3 Report

Revised manuscript does not address my 2nd review. Hopefully comments below will add clarity to my request.

109   My previous comment was to point out that “because considered inaccuracy” represents a grammatical error and therefore does not make sense. I presume that the authors are trying to say that the V-slope method is considered to provide a more accurate representation of VT than RER< 1.0.

My other comments regarding peak  RER relate to patient effort. As the authors are likely aware, RER < 1.1 is generally regarded as indicating excellent and/or maximal effort. This has implications for other peak measures. In cases where peak RER> 1.1 it could be argued that any group and/or test differences may  be attributed to low effort rather than pathology, treatmen,t or any other independent variable of interest.

The authors indicate “that the mean respiratory exchange ratio (RER; VCO2/VO2) of the last 15 seconds was calculated.” (line 101) However, I cannot find the actual values reported, nor any group x test analysis of peak RER. The authors do not follow the convention of providing a table of means and SD’s for all dependent variable,  but other peak values can be gleaned from 3.4. Figures and tables.

I want to know whether peak RER values met criteria for max effort, whether or not there were any group and or test differences and potential implications. Any incidence of peak values for RER> 1.1 represent possible study limitations and should be addressed.

Please run grammar checker/review again to further improve readability.

Author Response

Please see the document.
